# Antenatal depression and its association with adverse birth outcomes in low and middle-income countries: A systematic review and meta-analysis

**Abel Fekadu Dadi**[1,2]*, **Emma R. Miller**[2], **Lillian Mwanri**[2]

**1** Institute of Public Health, College of Medicine and Health Sciences, University of Gondar, Gondar, Ethiopia, **2** College of Medicine and Public Health, Flinders University, Health Sciences Building, Adelaide, South Australia

* Fekten@yahoo.com

**Data Availability Statement:** All relevant data are within the manuscript and its Supporting Information files.

**Funding:** Not funded

## Abstract

### Background

Depression in pregnancy (antenatal depression) in many low and middle-income countries is not well documented and has not been given priority for intervention due to competing urgencies and the belief that it does not immediately cause fatalities, which mainly emanated from lack of comprehensive research on the area. To fill this research gap, this systematic review was conducted to investigate the burden of antenatal depression and its consequences on birth outcomes in low- and middle-income countries.

### Methods

We systematically searched the databases: CINHAL, MEDLINE, EMCare, PubMed, PSyc Info, Psychiatry online, and Scopus for studies conducted in low and middle-income countries about antenatal depression and its association with adverse birth outcomes. We have included observational studies (case control, cross-sectional and cohort studies), written in English-language, scored in the range of "good quality" on the Newcastle Ottawa Scale (NOS), and were published between January 1, 2007 and December 31, 2017. Studies were excluded if a standardized approach was not used to measure main outcomes, they were conducted on restricted (high risk) populations, or had fair to poor quality score on NOS. We used Higgins and Egger's to test for heterogeneity and publication bias. Primary estimates were pooled using a random effect meta-analysis. The study protocol was registered in PROSPERO with protocol number CRD42017082624.

### Result

We included 64 studies (with 44, 035 women) on antenatal depression and nine studies (with 5,540 women) on adverse birth outcomes. Antenatal depression was higher in the lower-income countries (Pooled Prevalence (PP) = 34.0%; 95%CI: 33.1%-34.9%) compared to the middle-income countries (PP = 22.7%, 95%CI: 20.1%-25.2%) and increased

**Competing interests:** The authors have declared that no competing interest exist.

**Abbreviations:** CES-D 20, Center for Epidemiologic Studies depression scale 20; DM, Diabetic Mellitus; DSM-IV, Diagnostic and Statistical Manual of mental disorders version 4; EPDS, Edinburgh Postnatal Depression Scale; HIV, Human Immune Deficiency; IPV, Intimate Partner Violence; LAMICS, Low and Middle Income Countries; NOS, New Castle Ottawa Scale; OR, Odds Ratio; PDQ, Pitt Depression Questionnaire; PHQ-9, Patient Health Questiner-9; PND, Postnatal Depression; POR, Pooled Odds Ratio; PRISMA, Preferred Reporting Items for Systematic Review and Meta-analysis; RCT, Randomized Control Trial; RR, Relative Risk; SRQ-20, Self Reporting Questioner; WHO, World Health Organization.

over the three trimesters. Pregnant women with a history of economic difficulties, poor marital relationships, common mental disorders, poor social support, bad obstetric history, and exposure to violence were more likely to report antenatal depression. The risk of having preterm birth (2.41; 1.47–3.56) and low birth weight (1.66; 1.06–2.61) was higher in depressed mothers compared to mothers without depression.

## Conclusions

Antenatal depression was higher in low-income countries than in middle-income countries and was found to be a risk factor for low birth weight and preterm births. The economic, maternal, and psychosocial risk factors were responsible for the occurrence of antenatal depression. While there could be competing priority agenda to juggle for health policy-makers in low-income countries, interventions for antenatal depression should be reprioritized as vitally important in order to prevent the poor maternal and perinatal outcomes identified in this review.

## Introduction

Depression is a common mental health disorder worldwide, which can manifest as a depressed mood, feeling of guilt, loss of interest, low self-esteem, difficulty in getting adequate sleep, and lack of concentration in everyday life [1]. Globally, more than 300 million peoples of all age suffer from depression [1] with much higher prevalence in the African (9%) and South-east Asian (27%) regions [2, 3]. By 2030, depression is predicted to be the second and third leading cause of disease burden in developing and low-income countries, respectively [4]. Depression prevalence is higher in pregnant populations relative to general female populations, often due to hormonal changes during pregnancy [5]. A systematic review of studies conducted in developed and low-income countries reported an antenatal depression prevalence in the range of 5% to 30% [6–8] and 15.6% to 31.1%) [9–11], respectively. These estimates varied according to ethnicity, history of miscarriage, medically assisted pregnancy, ambivalent attitude about the current pregnancy, and socioeconomic condition of the women [6–8].

Maternal depression could affect household income, productivity, child development [12], and quality of life [13]. Pregnant women with depression can produce a high level of stress hormones such as cortisol that can subsequently affect fetal growth [14] and brain development [15, 16]. Depression during pregnancy has been reported as a risk factor for low birth weight [17] and preterm births [18–22] and may also affect the child stress coping ability in later life [23]. In contrast, some other studies have reported a lack of association between antenatal depression and adverse birth outcomes [24–28].

Depression manifests in different ways during pregnancy and the postnatal period [29], which could challenge its identification and treatment. One study reported a triadic pattern of depression during pregnancy including an increase during the first few weeks of pregnancy, a decrease mid-way during the pregnancy and another increase again after the final weeks of pregnancy [30]. A number of systematic reviews have been conducted on maternal mental health and its effect on birth outcomes. However, these were not specific to depression [9, 31, 32], did not focus on low and middle-income countries (where the problem is thought to be high) [7], and did not evaluate if there is any relationship between antenatal depression and risk of adverse birth outcomes [33]. Therefore, the current systematic review and meta-analysis

was conducted to explore the burden of antenatal depression, its risk factors and its association with adverse birth outcomes in low and middle-income countries.

## Methods

### Search strategy

CINHAL, MEDLINE, EMCare, PubMed, Psych INFO and, Psychiatry online, and Scopus data bases were systematically searched for the following key terms: Pregnan*, antenat*, depression, clinical depression, depressed mood, major depressive disorder, depressive symptom, adverse birth outcomes, stillbirth, preterm birth, and low birth weight. Example of full electronic search strategy in PubMed.

> Search ((((((Pregnant mothers*) OR (antenatal mothers*) OR (pregnant women*) OR (antenatal period*) OR pregnancy* OR (antepartum women*)) AND ((depression* OR (clinical depression*) OR (depressed mood*) OR(major depressive disorder*) OR (depressive symptom*) OR (psychological morbidity*) OR (major depression*) OR (unipolar depression*)) AND((exposure* OR (risk factor*) OR correlates* OR (associated factors*) OR predictors*) AND (((cross sectional*) OR (crosssectional*) OR survey* OR(case control*) OR (nested case control*)) Sort by: Publication Date Filters: Publication date from 2007/01/01 to 2017/12/31; Humans; English; MEDLINE; Field: Title/Abstract

### Eligibility criteria

We included observational studies (case-control, cross-sectional, and follow up studies) that were conducted in low and middle-income countries, written in the English language and published between January 1, 2007 and December 31, 2017. The following were the other criteria for studies to be selected for the review: depression was measured using validated screening tools; low birth weight was measured and classified as birth weight less than 2500grams; preterm birth was studied and defined as birth occurring before 37 complete weeks of gestation. Studies were excluded if a standardized approach was not used to measure main outcomes, conducted on restricted (high risk) populations such as studies conducted in refugee camps, conducted following certain disasters, conducted on mothers living with HIV/TB, restricted studies such as those exclusively conducted on first time mothers, women with high complication during pregnancy, grey literatures or had fair to poor quality score on NOS. The study inclusion, exclusion and reason for exclusion is presented in Fig 1 and page 11 of the S1 File.

### Study quality assessment

Identified studies were exported to Endnote version 7 and duplicates were removed. Two independent reviewers (AFD & BAD) conducted a full-text quality review. Disagreement between the two reviewers was found to be very low (1.5%) and they resolved this through discussion. The Newcastle-Ottawa Scale (NOS) [34, 35] for observational studies was used to assess the quality and risk of bias in included studies. The NOS includes 3 categorical criteria with a maximum score of 10 points: "selection" which accounts a maximum of 5 points, "comparability" which accounts a maximum of 2 points, and "outcome" which accounts a maximum of 3 points. The quality of each study was rated using the following scoring algorithm: ≥7 points was considered as "good" quality study, 2 to 6 points was considered as "fair" quality study, and ≤ 1 point was considered as "poor" quality study. Only studies of good quality

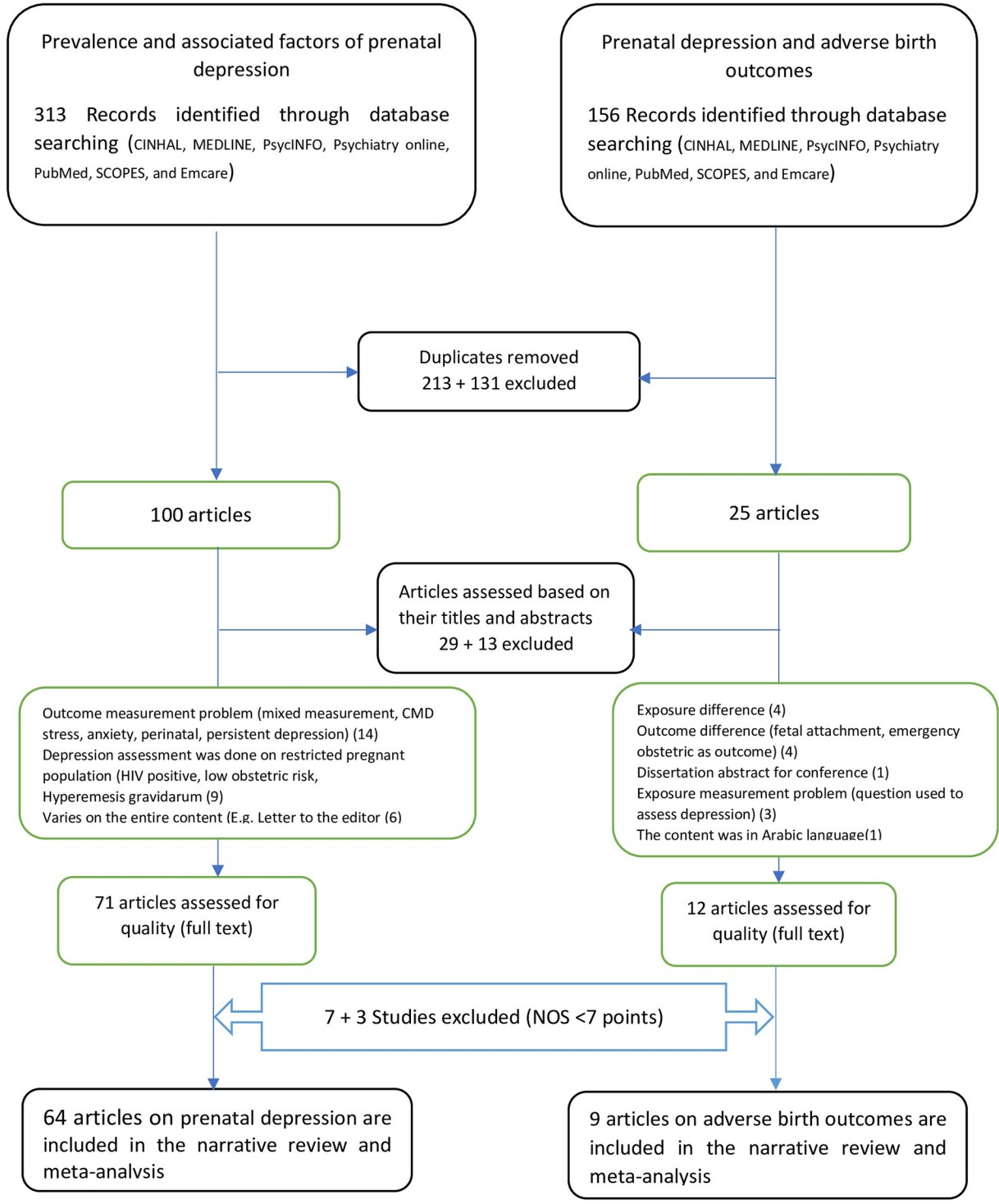

**Fig 1. Flow chart of study inclusion for systematic review and meta-analysis of antenatal depression and its effect on birth outcomes.**

(NOS score ≥7 points) were included in this systematic review and meta-analysis. The scoring of each quality assessment component for each study was presented in a table. (See in S1 File) This systematic review and meta-analysis was based on the Meta-analysis Of Observational Studies in Epidemiology (MOOSE)[36] statement.

## Data extraction

The following data from studies with good quality [NOS score ≥7 points] was extracted using a structured data extraction form and summarized in table format: Name of authors, year of publication, name of country in which the study was conducted, country income category, study design, sample size, type of screening tool used to identify depression and its cut of value, and the estimates (prevalences of antenatal depression with their confidence intervals and odds ratios with their confidence intervals for risk factors). (Tables 1 and 4)

## Data synthesis

The data synthesis was separately conducted for antenatal depression and birth outcomes. Meta-analysis of proportions for antenatal depression, odds ratios for factors associated with antenatal depression, and relative risks for reporting adverse birth outcomes were calculated after log-transforming the estimates from primary studies. If multiple outcomes were reported in a single study, each outcome was analysed independently.

## Risk of bias and adjustment

Funnel plot and Egger's regression test were conducted to check for the presence of potential publication bias [37, 38]. In the presence of publication bias, an estimate from Trim and Fill analysis was reported [39]. Galbraith plot [40] and Higgins test [41] were used to explore the presence of heterogeneity. Sub-analyses was conducted according to the identified sources of heterogeneity and the effect size from the random effect models was calculated [42]. Sensitivity analyses was also been conducted. All analysis was conducted in Stata 14 [43].

## Protocol registration

This review was registered in PROSPERO with a protocol number CRD42017082624. Available from: https://www.crd.york.ac.uk/prospero/display_record.php?ID=CRD42017082624.

# Results

Our search strategy identified 313 records for antenatal depression and 156 records for the effect of antenatal depression on birth outcomes. After duplicates were removed and preliminary screening of the titles and abstracts, 83 articles fulfilled a criterion for full-text review and quality assessment. Finally, 73 articles were assessed as good quality and included in the systematic review and meta-analysis. From these articles, 64 were conducted on antenatal depression and 9 articles were conducted to investigate the effect of antenatal depression on birth outcomes. (Fig 1)

## Antenatal depression prevalence and its associated factors

Among 64 articles conducted on antenatal depression, 49 (77%) were conducted in middle-income countries, 15 (23%) studies were conducted in low-income countries, and 46 (72%) were health institutional based studies. Half of the studies (32) investigated antenatal depression at all stages of pregnancy while 20 (31%) of the studies investigated antenatal depression during the last trimester of pregnancy. A relatively large number of studies, 20 (31%), were published in the year 2015–2016 and the majority of the studies, 34 (53%), used Edinburgh Postnatal Depression Scale (EPDS) for screening antenatal depression. (Table 1)

Because of a high heterogeneity index ($I^2$ = 96.7%, P<0.001) among included antenatal depression studies, we were unable to pool all the estimates but we conducted a sub-analysis based on the following characteristics: year of publication, country income category, study

**Table 1. Summary of studies conducted on antenatal depression in low and middle-income countries, (N = 64, in the year 2007–2017).**

| | Author, P. year | Country by income | Study setting | Study design | Sample size | Trimester screened | Tool used for screening | Prevalence |
|---|---|---|---|---|---|---|---|---|
| 1. | Adewuya, A. O. et al 2007[44] | Middle | HI | cross sectional | 180 | 3 | DSM-IV | 8.3% |
| 2. | Esimai, O. et al 2008[45] | Middle | HI | cross sectional | 195 | 1,2,3 | HADS | 10.8% |
| 3. | Gausia k et al, 2009[46] | Middle | Community | cross sectional | 361 | 3 | EPDS | 33% |
| 4. | Luna Matos M.L et al, 2009[47] | Middle | HI | cross sectional | 222 | 1,2,3 | EPDS | 40.1% |
| 5. | Mitsuhiro SS et al 2009[48] | Middle | HI | cross sectional | 1000 | 1,2,3 | CESD-10 | 12.9% |
| 6. | Pereira PK et al 2009[49] | Middle | HI | cross sectional | 331 | 3 | CESD-10 | 14.2% |
| 7. | Pottinger AM et al 2009[50] | Middle | HI | Longitudinal | 452 | 1,2,3 | EPDS | 25% |
| 8. | Golbasi Z et al 2009[51] | Middle | Community | cross sectional | 258 | 1,2,3 | EPDS | 27.5% |
| 9. | Silva RA et al 2010[52] | Middle | HI | cross sectional | 1264 | 1,2,3 | EPDS | 21.1% |
| 10. | Kaaya SF et al 2010[53] | Low | HI | cross sectional | 560 | 2 | HSC | 39.5% |
| 11. | Mohammad KI et al, 2011[54] | Low | HI | cross sectional | 353 | 1,2,3 | EPDS | 67.2% |
| 12. | Nasreen HE et al, 2011[55] | Low | Community | cross sectional | 720 | 3 | EPDS | 18.3% |
| 13. | Li J et al, 2011[13] | Middle | HI | cross sectional | 454 | 1,2,3 | EPDS | 39.0% |
| 14. | Lau Y et al 2011[56] | Middle | HI | cross sectional | 1609 | 2 | EPDS | 35.9% |
| 15. | Senturk V et al 2011[57] | Middle | HI | cross sectional | 971 | 3 | EPDS | 33.1% |
| 16. | Faisal-Cury A et al 2012[58] | Middle | HI | cross sectional | 312 | 2 | BDI | 21.1% |
| 17. | Melo Jr et al 2012[59] | Middle | HI | cross sectional | 600 | 3 | EPDS | 24.3% |
| 18. | Hartley M et al, 2011[60] | Middle | Community | cross sectional | 1062 | 1,2,3 | EPDS | 39.0% |
| 19. | Rochat TG et al, 2011[61] | Middle | HI | cross sectional | 109 | 2 | DSM-IV | 47% |
| 20. | Ajinkya s et al, 2012[62] | Middle | HI | cross sectional | 185 | 1,2,3 | BDI | 9.2% |
| 21. | Fisher J et al, 2012[63] | Middle | Community | Longitudinal | 419 | 1 | EPDS | 22.4% |
| 22. | Fisher J et al, 2012[63] | Middle | Community | Longitudinal | 419 | 3 | EPDS | 10.7% |
| 23. | Fisher J et al, 2012[63] | Middle | Community | Longitudinal | 419 | 1,2,3 | EPDS | 17.4% |
| 24. | Silva R et al, 2012[64] | Middle | Community | cross sectional | 1264 | 1,2,3 | EPDS | 20.5% |
| 25. | Lara MA et al, 2012[65] | Middle | Community | cross sectional | 250 | 1,2,3 | CESD-10 | 16.2% |
| 26. | Manikkam L et al, 2012[66] | Middle | HI | cross sectional | 387 | 3 | EPDS | 38.5% |
| 27. | Fadzil A et al, 2013[67] | Middle | HI | cross sectional | 175 | 1,2,3 | HADS | 10.3% |
| 28. | Jeong H et al, 2013[68] | Middle | Community | cross sectional | 1262 | 1,2,3 | EPDS | 20.2% |
| 29. | Bindt C et al 2013[69] | Middle | HI | Longitudinal | 719 | 3 | PHQ | 28.9% |
| 30. | Dibaba Y et al 2013[70] | Low | Community | cross sectional | 627 | 3 | EPDS | 19.9% |
| 31. | Gemta A et al 2013[71] | Low | HI | cross sectional | 660 | 1,2,3 | EPDS | 25.6% |
| 32. | Guo N et al 2013[72] | Middle, cot devoir | HI | Longitudinal | 654 | 3 | PHQ | 26.3% |
| 33. | Guo N et al 2013 | Middle, Ghana | HI | Longitudinal | 654 | 3 | PHQ | 28.3% |
| 34. | Dmitrovic BK et al, 2014[73] | Middle | HI | cross sectional | 212 | 3 | EPDS | 21.7% |
| 35. | Abujilban SA et al 2014[74] | Low | HI | cross sectional | 218 | 3 | EPDS | 57% |
| 36. | Actas S et al 2014[75] | Middle | HI | cross sectional | 266 | 1,2,3 | BDI | 18.8% |
| 37. | Stewart RS et al, 2014[76] | Low | HI | cross sectional | 583 | 2 | SRQ | 21.1% |
| 38. | Weobong B et al 2014[77] | Middle | Community | Longitudinal | 2086 | 1 | SRQ | 9.9% |
| 39. | Waqas A et al 2015[78] | Middle | HI | cross sectional | 289 | 3 | HADS | 31.8% |
| 40. | Barrios Y et al 2015[79] | Middle | HI | Longitudinal | 1521 | 1 | PHQ | 29.1% |
| 41. | de Oliveira F et al 2015[80] | Middle | HI | cross sectional | 358 | 3 | EPDS | 28.2% |
| 42. | Abdelhai R et al 2015[81] | Middle | HI | cross sectional | 376 | 1,2,3 | HADS | 10.4% |
| **43.** | Mahenge B et al 2015[82] | Low | HI | cross sectional | 1180 | 1,2,3 | HSC | 78.2% |
| 44. | Rwakarema M et al; 2015[83] | Low | HI | cross sectional | 397 | 1,2,3 | EPDS | 33.8% |
| 45. | Heyningen T et al 2015[84] | Middle | HI | cross sectional | 376 | 1,2,3 | CIS-R | 22.0% |
| 46. | Biratu A et al 2015[85] | Low | HI | cross sectional | 393 | 1,2,3 | EPDS | 24.9% |
| 47. | Bavle A et al 2016[86] | Middle | HI | cross sectional | 318 | 1,2,3 | EPDS | 12.3% |

*(Continued)*

**Table 1.** (Continued)

| | Author, P. year | Country by income | Study setting | Study design | Sample size | Trimester screened | Tool used for screening | Prevalence |
|---|---|---|---|---|---|---|---|---|
| 48. | George C et al 2016[87] | Middle | Community | cross sectional | 202 | 1,2,3 | CIS-R | 16.3% |
| 49. | Moshki et al 2016[88] | Middle | HI | cross sectional | 208 | 3 | EPDS | 37.0% |
| 50. | Padmapriya N et al 2016[89] | Middle | Community | Longitudinal | 1144 | 1 | EPDS | 7.3% |
| 51. | Alvarado-EC et al 2016[90] | Middle | HI | cross sectional | 270 | 1,2,3 | EPDS | 37.4% |
| 52. | de Jesus Silva M et al 2016[91] | Middle | HI | cross sectional | 209 | 1,2,3 | HADS | 14.8% |
| 53. | de Moraes EV et al 2016[92] | Middle | HI | cross sectional | 375 | 1,2,3 | HADS | 40.8% |
| 54. | Malqvist M et al 2016[93] | Middle | Community | cross sectional | 1038 | 3 | EPDS | 22.7% |
| 55. | Thompson O et al 2016[94] | Middle | HI | cross sectional | 314 | 1,2,3 | EPDS | 24.5% |
| 56. | Ayele TA et al 2016[95] | Low | HI | cross sectional | 388 | 1,2,3 | BDI | 23.0% |
| 57. | Bisetegn TA et al 2016[96] | Low | Community | cross sectional | 527 | 1,2,3 | EPDS | 11.8% |
| 58. | Bitew T et al 2016[97] | Low | Community | cross sectional | 1311 | 2 | PHQ | 29.5% |
| 59. | Gelaye B et al 2017[98] | Middle | HI | cross sectional | 1298 | 2 | PHQ | 10.3% |
| 60. | Huanging H et al 2017[99] | Middle | HI | cross sectional | 4210 | 1,2,3 | HADS | 12.5% |
| 61. | Shidhaye P et al 2017[100] | Middle | HI | cross sectional | 302 | 1,2,3 | EPDS | 16.9% |
| 62. | Coll CVDN et al 2017[101] | Middle | Community | Longitudinal | 4130 | 2 | EPDS | 16.0% |
| 63. | Mossie Tb et al 2017[102] | Low | HI | cross sectional | 196 | 1,2,3 | BDI | 31.1% |
| 64. | Sahile MA et al 2017[103] | Low | HI | cross sectional | 233 | 3 | BDI | 31.2% |

**HSC**: Hopkins Symptom Checklist **CIS-R**: Clinical Interview Schedule–Revise **HI**: Health Institution **BDI**: Beck Depression Inventory

**EPDS**: Edinburgh Postnatal Depression scale **HADS**: Hospital Anxiety and Depression Scale **CIS-R**: Clinical Interview Schedule Revised

**SRQ**: Self Reporting Questionnaire **CESD-10**: Center for Epidemiological Studies Depression Scale **DSM-V**: Diagnostic and Stastical Manual of Mental Disorder **PHQ**: Patient Health Questionnaire

setting, sample size, tools used for screening, and time of pregnancy at which the screening has been conducted. As the Egger's test for publication bias was significant (P<0.001), Tweedie's and Duval's trim and fill analysis was used to report the final effect size under the random effect model. The pooled odds ratio was not affected when individual studies were omitted during the sensitivity analysis. (See in S1 File)

High prevalence of antenatal depression was estimated in the year 2011–2012, (pooled prevalence (PP), 95%CI: 28.6%; 22.3%-34.8%), followed by the year 2015–2016, (PP = 26.8%, 95% CI: 17.8%-35.8%). The antenatal depression was higher in low-income countries (PP = 34.1%, 95%CI: 22.7%-45.6%) and in health institution-based studies (PP = 27.6%, 95%CI: 23%-32.3%) as compared to high-income countries and community-based studies, respectively. Antenatal depression increased over the three trimesters; 17.1% (95%CI: 7.7%-26.5%) in the first trimester, 27.1% (95%CI: 19.7%-34.6%) in the second trimester, and 28.9% (95%CI: 23.7%-34.1%) in the third trimester. The antenatal depression prevalence was estimated to be higher among the studies with sample less than 600 participants (PP = 25.7%; 95%CI: 22.1%-29.5%) and those that used the Hopkins Symptom Checklist for screening depression (PP = 58.9%, 95%CI; 21%-96.8%). (Table 2)

We have also summarized and pooled the effect size for reported risk factors under relatively homogeneous groups. Accordingly, bad obstetric history (Pooled Odds Ratio (POR) = 2.01; 95%CI: 1.67, 2.42) in 16 studies and economic difficulties in 14 studies (POR = 2.03; 95% CI: 1.63, 2.53) were significantly associated with increased risk of antenatal depression. Similarly, having poor social support (POR = 1.77; 95%CI: 1.49, 2.10) and history of common mental disorders (POR = 3.27; 95%CI: 2.47, 4.33) were increased the risk of antenatal depression in 13 studies. Moreover, having a history of violence in 11 studies (POR = 2.99; 95%CI: 2.20,

**Table 2. Sub-analysis of antenatal depression prevalence in the low- and middle-income countries (N = 64, 2007–2017), (random effect model).**

| Variables of sub-analysis | Number of studies (%) | Sample size | Pooled prevalence; 95%CI |
|---|---|---|---|
| Year of publication | | | |
| 2007–2008 | 2(3.13) | 375 | 9.46 (6.5–12.5) |
| 2009–2010 | 8(12.50) | 4,448 | 26.4(19.5–33.4) |
| 2011–2012 | 16(25.00) | 9,533 | 28.6(22.3–34.8) |
| 2013–2014 | 12(18.75) | 8,116 | 23.8(18.3–29.3) |
| 2015–2016 | 20(31.25) | 11,194 | 26.8(17.8–35.8) |
| 2017 | 6(9.38) | 10,369 | 18.2(14.6–21.8) |
| Income of the country | | | |
| Low income | 15(23.44) | 8346 | 34.1(22.7–45.6) |
| Middle income | 49(76.56) | 35,689 | 22.7(20.1–25.2) |
| Study setting | | | |
| Health institution | 46(71.88) | 26536 | 27.6(22.9–32.3) |
| Community based | 18(28.13) | 17499 | 19.8(16.1–23.5) |
| Time of screening | | | |
| First trimester | 4(6.25) | 5170 | 17.1 (7.7–26.5) |
| Second trimester | 8(12.50) | 9912 | 27.1(19.7–34.6) |
| Third trimester | 20(31.25) | 9532 | 28.9(23.7–34.1) |
| All trimester | 32(50.00) | 19421 | 23.9(18.2–29.5) |
| Median sample size | | | |
| < = 600 | 42(65.62) | 13290 | 25.7(22.0–29.5) |
| >600 | 22(34.38) | 30745 | 24.8(18.9–30.6) |
| Tool used for screening depression | | | |
| EPDS | 34(53.13) | 23,612 | 27.2(23.5–30.8) |
| CIS-R | 2(3.13) | 578 | 19.3(13.8–24.9) |
| PHQ-9 | 6(9.38) | 6157 | 25.4(17.3–33.4) |
| SRQ-20 | 2(3.13) | 2669 | 15.4(4.4–26.4) |
| Hopkins symptom checklist | 2(3.13) | 1740 | 58.9(20.9–96.8) |
| BDI | 6(9.38) | 1580 | 22.2(15.7–28.6) |
| HADS | 7(10.94) | 5829 | 18.6(11.8–25.4) |
| DSM-IV | 2(3.13) | 289 | 27.4(10.5–65.3) |
| CESD-10 | 3(4.69) | 1581 | 13.6(11.9–15.3) |

**BDI**: Beck Depression Inventory **EPDS**: Edinburgh Postnatal Depression scale **HADS**: Hospital Anxiety and Depression Scale **CIS-R**: Clinical Interview Schedule Revised **SRQ**: Self Reporting Questionnaire **CESD-10**: Center for Epidemiological Studies Depression Scale **DSM-V**: Diagnostic and Stastical Manual of Mental Disorder

4.07), unsatisfied with relationship in 9 studies (POR = 2.18; 95%CI: 1.64, 2.90), and male gender preference in four studies (POR = 1.41; 95%CI: 2.97; 6.26) were the other factors associated with an increased risk of antenatal depression. (Table 3)

## Association of antenatal depression with adverse birth outcomes

From nine studies conducted to investigate the association of antenatal depression with adverse birth outcomes, six were from middle-income countries and community-based studies while half of them used the EPDS as a screening tool to measure depression. Almost all, 8 (90%) of the studies were prospective studies with a total sample of 5,540. The low birth weight was reported in seven studies but was found to be significantly associated with antenatal

**Table 3. Risk factors associated with antenatal depression, a meta-analysis of studies in the low- and middle-income countries (N = 64, 2007–2017), (estimate from random effect model after trim and fill analysis).**

| Variable of sub-analysis | Number of studies | Sample size | POR, 95%CI | I², p-value |
|---|---|---|---|---|
| **Poor obstetric history** (*history of adverse birth outcome, unwanted pregnancy, obstetric complications*) | 16 | 13450 | 2.01(1.67,2.42) | 81.7%, p = 0.137 |
| **Economic difficulties** | 14 | 11207 | 2.03(1.63,2.53) | 74.3%, p = 0.001 |
| **Poor social support** | 13 | 7372 | 1.77(1.49,2.10) | 85.7%, p = 0.001 |
| **History of CMD**(*depression, anxiety, stressful life events*) | 13 | 11799 | 3.27(2.47,4.33) | 89.9%, p = 0.001 |
| **History of all forms of violence** | 11 | 7428 | 2.99(2.20, 4.07) | 71.7%, p = 0.001 |
| **Unsatisfied marital condition** (*Unmarried, divorced, separated, shorter marital duration, polygamous*) | 9 | 7533 | 2.18(1.64,2.90) | 73.0%, p = 0.001 |
| **Male gender preference** (the family preferred male than girl) | 4 | 1135 | 2.97(1.41,6.26) | 88.2%, p = 0.001 |

depression in five of the studies. Similarly, two of four studies reported a significant association between antenatal depression and risk of preterm birth. (Table 4)

The risk of adverse birth outcomes (low birth weight or preterm birth) was 1. 59 times (95% CI: 1.34–2.92) higher among pregnant mothers who had signs of depression as relative to those did not. (Fig 2). Compared to LBW, the risk of PB was significantly higher among pregnant mothers with signs of depression (Pooled Relative Risk (PRR) = 2.41; 95%CI: 1.47–3.56). As the test for heterogeneity (I²; 81.1%, p = 0.0) and small study effect were significant (P<0.001), the final effect size was reported from Tweedie's and Duval's trim and fill analysis in the random effect model. (Figs 3 and 4) We did not find any influential study in our sensitivity analysis. (Fig 5)

**Table 4. Summary of studies conducted on the association of antenatal depression with adverse birth outcomes in the low and middle-income countries, (N = 9, in the year 2007–2017).**

| Author, Year | Country, income | Study setting | Study design | Sample size | Follow up start time | Tool used for screening | LBW (<2500gm) Estimate (RR/OR) | PB(<37weeks), Estimate (OR) |
|---|---|---|---|---|---|---|---|---|
| Rahman A et al, 2007[104] | Pakistan, Low | Community | Prospective cohort | 290 | 3rd | ICD-10 | 1.9;1.3–2.9) | |
| Nasreen HE et al 2010[105] | Bangladesh, Middle | Community | Prospective cohort | 720 | 2nd and 3rd | EPDS> = 10 | 2.24, 1.37–3.68 | |
| Niemi M et al, 2013[106] | Vietnam, Middle | Community | Prospective cohort | 334 | 3rd | EPDS > = 3 | 2.40;1.09–5.25 | 2.07, 1.2–3.56 |
| Sanchez SE et al, 2013[107] | Peru, Middle | HI | Case control | 959 | 3rd | PHQ-9> = 10 | | 3.67, 2.09–6.46 |
| Chang HY et al, 2014[26] | Korea, Low | HI | Prospective cohort | 691 | 3rd | CESD-10> = 10 | 1.66; 0.55–5.02 | |
| Husain N et al, 2014[27] | Pakistan, Low | Community | Prospective cohort | 763 | 3rd | EPDS > = 12 | 0.88; 0.73–1.06 | |
| Rao D et al, 2015 [108] | India, Middle | HI | Prospective cohort | 150 | 2nd & 3rd | PHQ-9> = 5 | | 3.3, 0.99–11.17 |
| Bindt C et al 2013[69] | Ghana, Middle | HI | Longitudinal, birth cohort | 719 | 3rd | PHQ-9 > = 10 | β = 52.2; 18.2–122.6 | 2.1, 0.8–5.6 |
| Wado WD et al 2014[109] | Ethiopia, Low | Community | Longitudinal, birth cohort | 537 | 2nd & 3rd | EPDS> = 13 | 1.77; 1.03–3.04 | |

**LBW**: Low Birth weight **HI**: Health Institutions **ICD-10**: International classification of Disease 10th **EPDS**: Edinburgh Postnatal Depression Scale **SRQ**: Self Reporting Questionnaire **CESD-10**: Center for Epidemiological Studies Depression Scale **DSM-V**: Diagnostic and Statical Manual of Mental Disorder **PHQ**: Patient Health Questionnaire

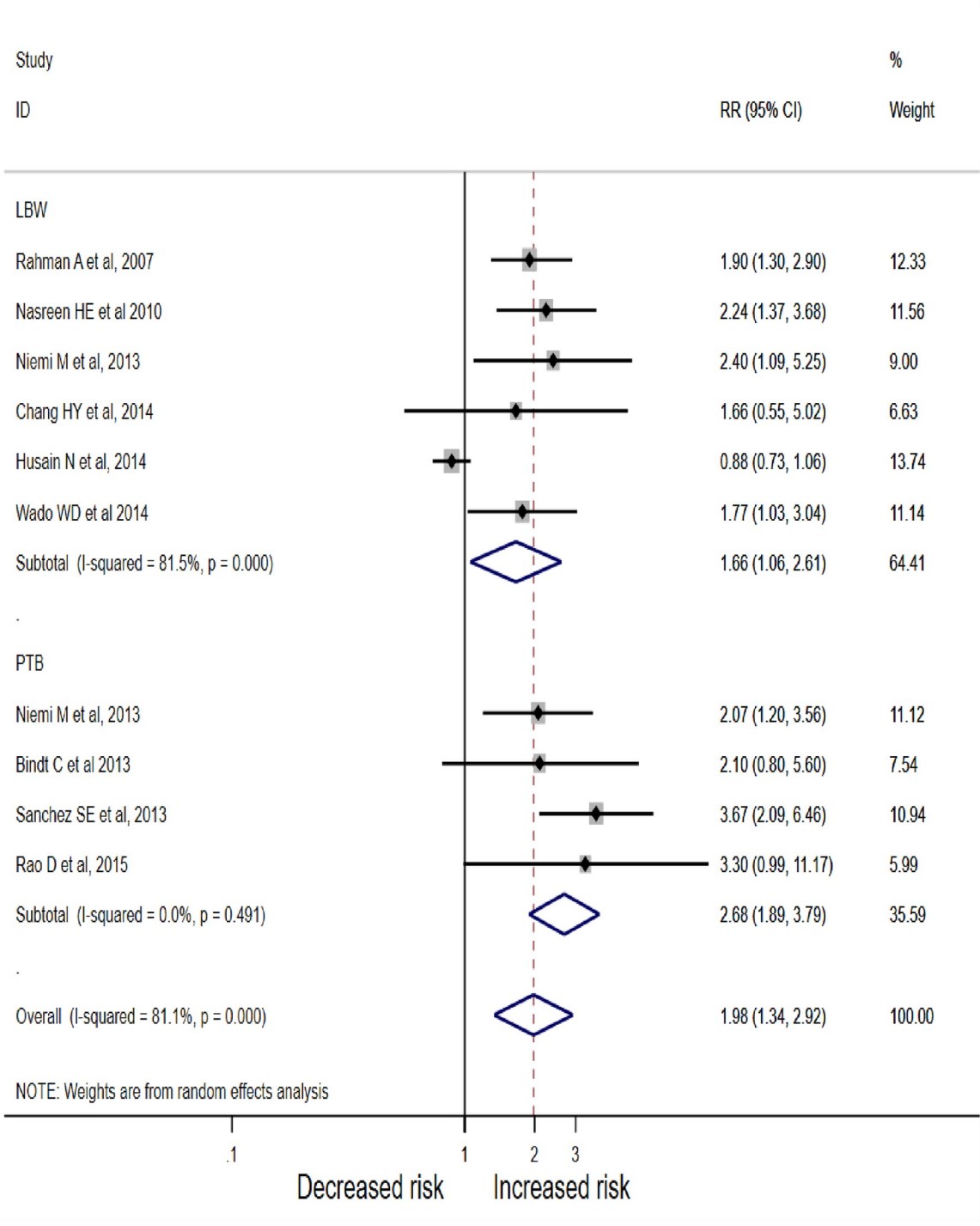

RR: relative risk, PTB: Preterm birth, LBW: Low birth weight

**Fig 2. Association between antenatal depression and adverse birth outcomes (N = 9, 2007–2017).**

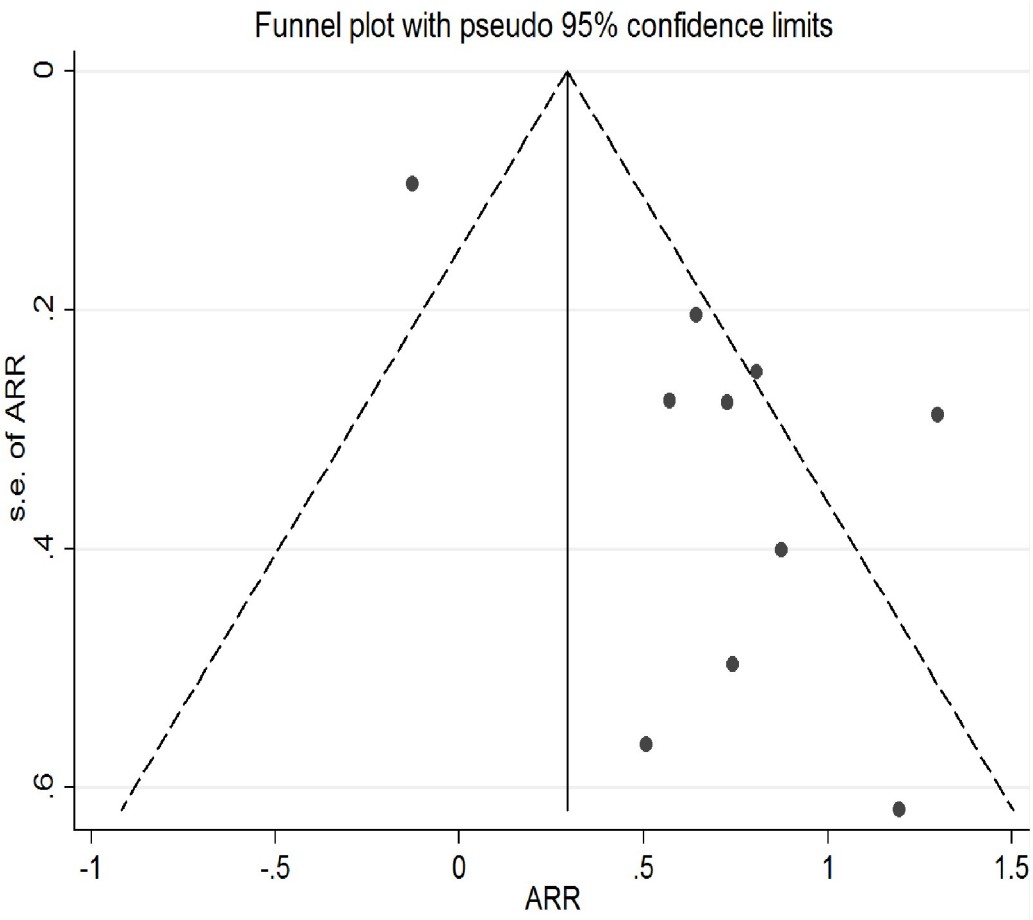

**Fig 3. Funnel plot before Tweedie's and Duval's trim and fill analysis.**

In the sub-analyses, relative to high income countries, the risk of adverse birth outcomes was significantly higher among mothers from middle-income countries (PRR = 2.51; 95%CI: 1.92–3.28), in health institution-based studies (PRR = 2.92; 95%CI: 1.92–4.43), and when depression commenced in the second trimester (PRR = 2.47; 95%CI: 1.76–3.46). The association between antenatal depression and adverse birth outcomes did not differ between studies in which pregnant mothers were clinically diagnosed with depression and were identified based on a self-reported scale of depression symptom. (Table 5)

## Discussion

This review has provided strong evidence for the burden of antenatal depression and its association with adverse birth outcomes in low and middle-income countries. To our knowledge, this review represents the first attempt to quantify this information and provides valuable impetus for the development of interventions aimed at addressing issue which has so far been neglected in the countries with greatest antenatal depression prevalence.

We found the prevalence of antenatal depression in the low-income countries was higher than that of middle-income countries and has increased from 9.5% in 2007 to 18.2% in 2017. The increase in prevalence over time might be attributed to the increase in number of studies on the topic as a result of the problem got more attention by researchers or the prevalence has

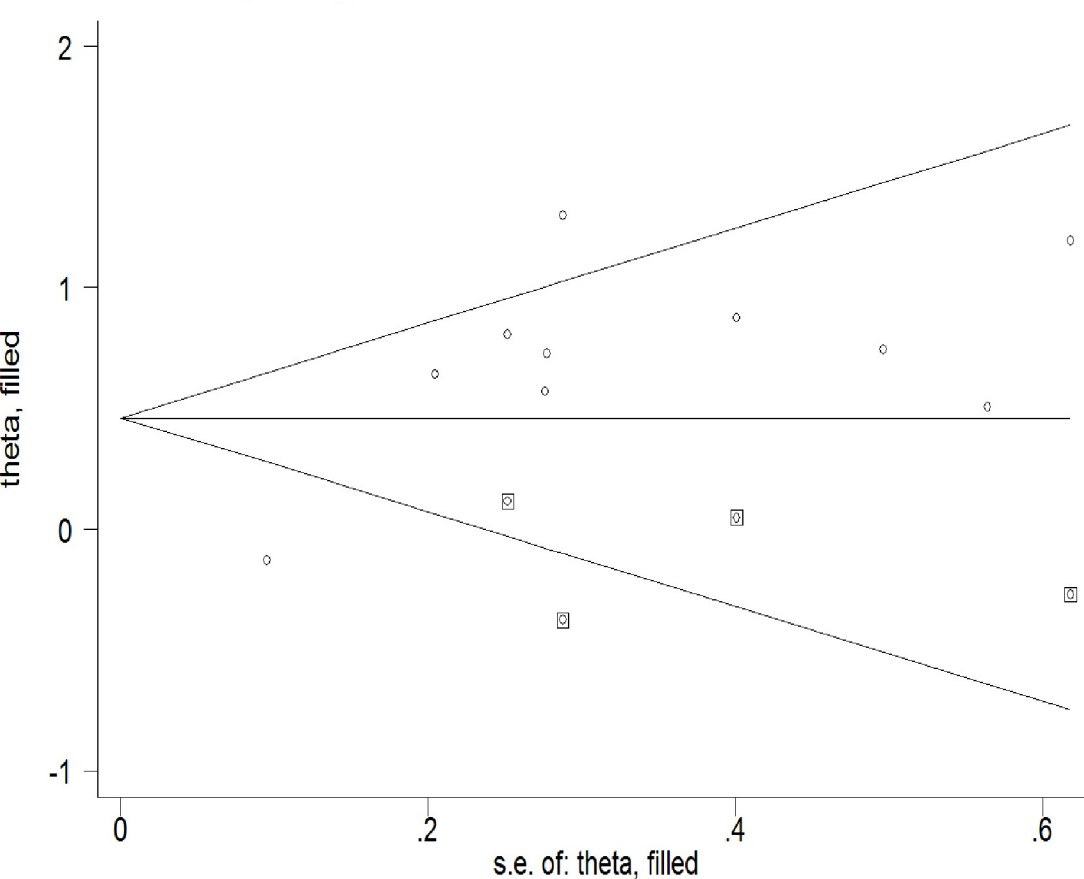

**Fig 4. Funnel plot after Tweedie's and Duval's trim and fill analysis (Filled by four studies).**

been increasing overtime because of low attention has been given for the problem by different countries. This could be also exemplified in the sub-analysis table in the result section of this article. The increase does, however, support the prediction that depression will become the third leading cause of disease burden in the low-income countries by 2030 [110, 111]. Consistent with previous reviews [10, 112], we found significantly higher antenatal depression prevalence in low-income countries relative to middle-income countries. This might be because depression has not previously been prioritized as an area for intervention relative to other problems during pregnancy [112]. It is also likely that risk factors associated with mental health disorders are more common in low-income countries [10].

We found that antenatal depression prevalence increases from the first to the third trimester of pregnancy, which contrast with the quadratic pattern (increase during the first trimester, drop in the second, and increase during the third trimester) noted elsewhere [30]. This might associate with the number of included studies during different trimesters, the pooled prevalence was high where large number of studies included and low where small studies were included as adjustment was not made on the number of studies. Consistent with our findings, one study reported an increased pattern of depression from the first to the second trimester due to increases in a range of risk factors during the three trimesters of pregnancy [113]. Further study could assist in prevention planning in identifying the appropriate timing and frequency of screening and intervention. Further investigation could also help to identify risk

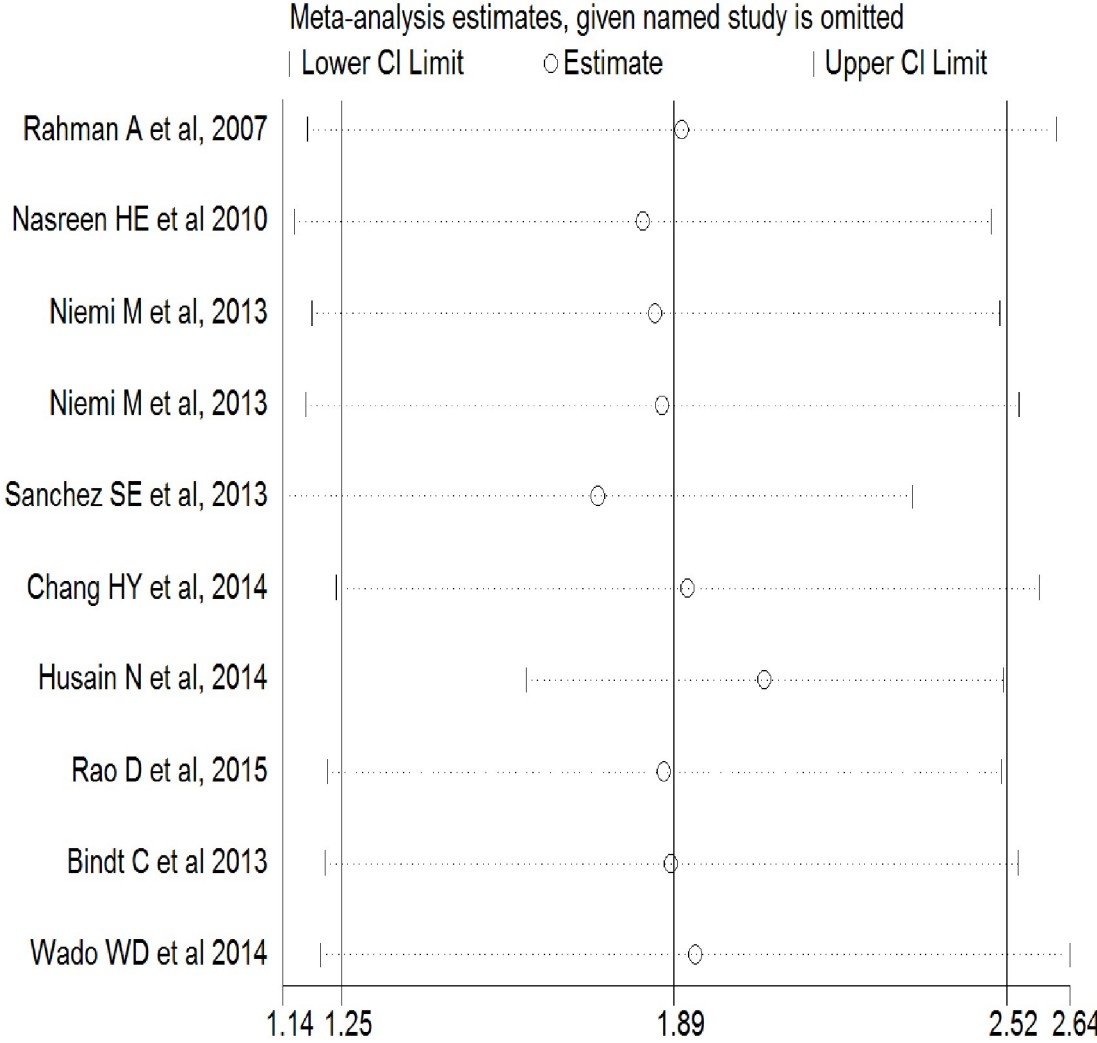

**Fig 5. Sensitivity analysis for studies on antenatal depression and its effect on adverse birth outcomes (N = 9, 2007–2017).**

factors that might change in the level of influence over the pregnancy, which could better target interventions across the three trimesters.

However, there are some methodological issues that might cause variation in estimations. For instance, we noted that institutional based studies reported higher prevalence relative to community-based studies and studies with smaller sample sizes reported higher prevalence relative to larger studies and these are mainly explained by the inherence limitations of cross-sectional studies. Estimate size also varied according to the tools used to measure depression. While the estimates from EPDS were the most consistent one with previous similar reviews, there was greater variation among estimates from studies using other tools [10, 114, 115].

We found that previous medical conditions (bad obstetric history and history of the previous episode of common mental disorders) and social or cultural factors (poor social support, financial difficulties, exposure to all forms of violence during pregnancy or childhood, unfavorable marital conditions, and male gender preference) were an important risk factors for antenatal depression.

**Table 5. Sub-analysis of the association of antenatal depression with adverse birth outcomes in the low- and middle-income countries (N = 9, in the year 2007–2017), (random effect model).**

| Variable of sub-analysis | Number of studies (%) | Sample size | Pooled RR; 95%CI | $I^2$, p-value |
|---|---|---|---|---|
| **Income of the country** | | | | |
| Low income | 4 | 2324 | 1.42(0.85,2.38) | 81.3%, p = 0.029 |
| Middle income | 5 | 3216 | 2.51(1.92,3.28) | 0.0%, p = 0.736 |
| **Study setting** | | | | |
| Health institution | 4 | 2519 | 2.92(1.92,4.43) | 0.0%, p = 0.607 |
| Community based | 5 | 3021 | 1.72(1.11,2.67) | 83.6%, p = 0.002 |
| **Time follow up started** | | | | |
| 2nd and 3rd trimester | 3 | 2366 | 2.47(1.76, 3.46) | 19.3%, p = 0.454 |
| Third trimester | 6 | 3174 | 1.66(1.04,2.66) | 78.9%%, p = 0.026 |
| **Tool used for depression screening** | | | | |
| EPDS | 4 | 2688 | 1.70(1.01,2.83) | 84.1%, p = 0.01 |
| PHQ-9 | 3 | 1828 | 3.20(2.04,5.04) | 0.0%, p = 0.633 |
| CESD-10/ICD-10 | 2 | 1024 | 1.87(1.28,2.73) | 0.0%, p = 0.843 |
| **Type of adverse birth outcome** | | | | |
| Low birth weight | 6 | 3712 | 1.66(1.06,2.61) | 81.5%%, p = 0.008 |
| Preterm birth | 4 | 2496 | 2.41(1.47,3.56) | 0.0%, p = 0.620 |
| **Sample size** | | | | |
| <350 | 3 | 1151 | 2.07(1.55,2.77) | 0.0%, p = 0.83 |
| > = 350 | 6 | 4389 | 1.84(1.05,3.25) | 85.9%, p = 0.001 |

Bad obstetric histories such as unwanted pregnancy, multiparity, history of miscarriage, still and preterm birth, and other un specified complications were reported in 15 studies. Having greater numbers of children might have considerable economic impact and stress and could be further exacerbated by an additional unwanted pregnancy [112]. History of miscarriage, still- and preterm birth (defined here as a negative obstetric history) may be associated with trauma and fear in relation to the current pregnancy outcome [9]. Other pregnancy complications such as hyperemesis gravidarum, hypertension, and diabetes mellitus could also pose additional stress on mothers [116].

Maternal or familial history of common mental disorders (such as depression, anxiety, stressful life event, and any other psychiatric issue) predicted the current depression episodes in 13 studies. This provides support for the familial and recurrent nature of depression and other mental health morbidities as pointed out by Shyn and Hamilton [117]. Concomitant exposure to stressful life events could also trigger the occurrence of depression by playing an additive role in the causal process [10, 118, 119].

Good social support [120] could positively affect the mother's stress coping ability by playing a buffering role in the causal model [10, 120, 121] means social support significantly reduces the risk of antenatal depression. Moreover, studies have shown a preventive effect of balanced nutritional interventions [122, 123] during pregnancy on antenatal depression. However, mothers in low income countries are living in economic pressure which also indirectly affect their adherence to proper nutrition during pregnancy. Exposure to sexual, physical, and emotional violence before or during pregnancy or history of childhood abuse was associated with an increased risk of antenatal depression. This is associated with disruption of neurobiological and stress response system through changing of brain structure and function [124–126] and the prevalence of such forms of violence in low and middle-income countries is known to be high [127].

Reduced relationship satisfaction with partners was reported as a risk factor for having depression symptoms. The risk of depression was also higher when the pregnant mothers are from a family that prefers male than a girl in the current pregnancy. This gender preference could directly affect maternal support and combined with a problem with partner relationship could brought maternal distress, loneliness and, ultimately, depression throughout the pregnancy [10, 119].

After accounting for publication bias, an exposure history of antenatal depression was associated with a 59% higher risk of adverse birth outcomes. In the sub-analysis according to a type of adverse birth outcomes, the risk of preterm birth was higher compared to the risk of low birth weight. A significant association between antenatal depression and low birth was also reported in two systematic reviews [33, 128], however, Kathleen et al reported absence of association between antenatal depression and adverse birth outcomes [129].

An increased preterm birth risk (1.4) among mothers with depression history was consistent with a meta-analysis published by Grigoriadis et al [130]. Similarly, a 1.2 times risk of preterm birth and 1.3 times risk of low birth weight was reported in a meta-analysis conducted by Grote, which is also in line with our finding [131]. More importantly, this review found that the association of antenatal depression on adverse birth outcomes was similar among studies that used clinical investigation and studies that used self-reported screening tool to identify pregnant mothers with depression.

The causal mechanisms between antenatal depression and adverse birth outcomes could be explained in multiple ways: (1) Depression may exert an influence on adverse birth outcomes via dysregulation of the Hypothalamic-Pituitary-Adrenocortical Axis [132] that stimulates the release of stress hormone such as cortisol, which could prevents adequate oxygen and nutrient flow to the fetus [129, 133]; (2) Antenatal depression might also disrupt immune system dysfunction that leads the mothers to develop different type of infections and, potentially, affects fetal growth [131]; (3) Depressed mothers may be more likely to smoke and drink while being less likely to attend medical care [134–136] and have poor appetite all of which can lead to malnourishment and impact on fetal development [137].

## Limitations

We included all available high-quality studies on antenatal depression and its effect on adverse birth outcomes, however, our estimation may still have been subject to measurement bias due to variation in diagnostic approaches among studies. Moreover, language restrictions might also introduce the risk of publication bias. Nonetheless, our analytical approach addressed heterogeneity and publication bias and provides some confidence in our estimates of the burden and consequences of antenatal depression in low and middle-income countries.

## Conclusion

We found that antenatal depression is highly prevalent and increases over the duration of pregnancy. We also noted increases in prevalence over the last ten years. Antenatal depression prevalence was found to be higher in low-income countries relative to middle-income countries. The current review has identified risk factors for pregnant mothers at higher risk of developing antenatal depression such as; bad obstetric history, previous episode of common mental disorders, poor social support and financial difficulties. Similarly, women reporting a history of exposure to violence (during pregnancy or earlier) and unsatisfactory relationships were more at risk of developing depression. A strong association between antenatal depression and adverse birth outcomes, which was not affected by method of depression identification in pregnant mothers, was also noted in the current review. While there could be competing

priority agenda to juggle for health policymakers in low-income countries, interventions for antenatal depression should be reprioritized as vitally important in order to prevent the poor maternal and perinatal outcomes identified in this review.

## Supporting information

**S1 File.**
(DOCX)

## Acknowledgments

Our heartfelt gratitude will go to Mr. Berihun Assefa Dachew for his contribution in quality assessment of the included studies.

## Author Contributions

**Conceptualization:** Abel Fekadu Dadi, Emma R. Miller, Lillian Mwanri.

**Data curation:** Abel Fekadu Dadi.

**Formal analysis:** Abel Fekadu Dadi.

**Methodology:** Abel Fekadu Dadi.

**Project administration:** Abel Fekadu Dadi.

**Supervision:** Emma R. Miller, Lillian Mwanri.

**Validation:** Emma R. Miller, Lillian Mwanri.

**Writing – original draft:** Abel Fekadu Dadi.

**Writing – review & editing:** Abel Fekadu Dadi, Emma R. Miller, Lillian Mwanri.

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
