## [Decision Letter · Decision Letter 0]

7 Oct 2019

PONE-D-19-25076

Prenatal Depression and its Effect on Birth Outcomes in Low and Middle-income Countries: A Systematic Review and Meta-analysis

PLOS ONE

Dear Mr Fekadu,

Thank you for submitting your manuscript to PLOS ONE. After careful consideration, we feel that it has merit but does not fully meet PLOS ONE’s publication criteria as it currently stands. Therefore, we invite you to submit a revised version of the manuscript that addresses the points raised during the review process.

We would appreciate receiving your revised manuscript by Nov 21 2019 11:59PM. To enhance the reproducibility of your results, we recommend that if applicable you deposit your laboratory protocols in protocols.io, where a protocol can be assigned its own identifier (DOI) such that it can be cited independently in the future. For instructions see: http://journals.plos.org/plosone/s/submission-guidelines#loc-laboratory-protocols

We look forward to receiving your revised manuscript.

Kind regards,

Animut Alebel, BSc, MSc

Academic Editor

PLOS ONE

Journal Requirements:

1. Please provide any updates you might have since the original search was performed in XXX, or please provide the rational for ending your search at that time.

Reviewers' comments:

Reviewer's Responses to Questions

**Comments to the Author**

1. Is the manuscript technically sound, and do the data support the conclusions?

Reviewer #1: Yes

Reviewer #2: Partly

2. Has the statistical analysis been performed appropriately and rigorously? 

Reviewer #1: Yes

Reviewer #2: Yes

3. Have the authors made all data underlying the findings in their manuscript fully available?

Reviewer #1: Yes

Reviewer #2: Yes

4. Is the manuscript presented in an intelligible fashion and written in standard English?

Reviewer #1: Yes

Reviewer #2: No

5. Review Comments to the Author

Reviewer #1: Thank you for the opportunity to review this interesting and informative piece. I found, at times, there were minor edits required especially in the abstract. Overall I found the paper well considered, referenced, and articulated. I strongly recommend the authors follow a standard format for their reference list - it is a mixture of different formats which is distracting and indicates an editing issue.

Personally i would appreciate a visual (such as a flow chart) of article identification, fit with criteria, etc.

Again, I appreciate the quality of your contribution.

Reviewer #2: Manuscript Number: PONE-D-19-25076

Prenatal Depression and its Effect on Birth Outcomes in Low and Middle-income Countries: A Systematic Review and Meta-analysis

Thank you very much for the chance you gave me to review the manuscript detailed above.

The authors have rigorously conducted a systematic review and met analysis that estimates the pooled prevalence of prenatal depression and also indicates its effect on birth outcome in low and middle income countries. In addition, the authors comes to address one of the top priority public health challenges. Maternal and their birth outcome one of the most important indicators that indicates the health status of a given countries.

Overall it is of good quality and relevant especially to the considered study setting.

However the authors need to revise the manuscript to suit and satisfy the reader, scientific communities like typos,

Thanks for the work. My detailed comments are given below

Abstract

The authors loosely described the gap of the research. The authors also need to describe the statistical issues in this section.

Methods:

The authors did not follow the PRISMA checklist for this systematic review. I recommend them to revise based on this systematic review is based on the Preferred Reporting Items of Systematic Reviews and Meta-Analysis (PRISMA) checklist guidelines to ensure scientific rigor. This systematic review lacks format especially in the method section. The authors also did not mention several subtitled for what they are mentioned in the method section. (Information sources); Authors did not include other major databases such Hinari?

Authors did not report anything about the grey literature that also needs to be explained here.

Newcastle-Ottawa Scale is over simplified, and authors should also consider NIH scale for observational studies as it is more detailed one to assess quality.

Result section

Throughout the manuscript the authors need to use the journal preparation guideline while citing the figures.

Page 5, paragraph 2 and 1 line, two independent reviewers … the authors need to mention who these authors are? Apply this comment also throughout the manuscript. On the same paragraph, line the authors considered good quality articles according to Newcastle Ottawa Scale greater than 7. What is your reference to use 7 as a cut-off point?

Page 6, the authors described that 46% of the included studies were health institutional based studies while the rest are community. So, why authors merge the two different setting studies? Are they similar population? How do you see the PICO criteria?

The authors described that they performed a subgroup analysis based on the publication year. 1) What is the advantage of reporting subgroup analysis based on publication year? I recommend the authors to report study period than publication year. Some articles published within one year of its done while others last 2/3/4/5 and more year that means a study conducted in 2010 may be published in 2013. Similarly what about publication between 2012 to 2014? 2) What is the base for categorizing publication year as 2011-2012 and 2015-2016? I need strong justification on this.

Discussion

Generally the authors need to improve more this section. All the articles that present the results of an investigation have to end in a section of discussion of results and conclusions. Therefore better discussion sounds reader and increase the quality of the study. These sections of the article should not be confused as they answer three clear questions, what was found? For the Results section, what is the meaning of what I found? When we are referring to the Discussion section and what are the most important findings of our work?

Specifically

Page #9, the authors mentioned that one of the reason for the increased postnatal depression is number of studies conducted in the setting. How could this be a reason for the increment?

Page #9, first line of last paragraph, the authors reported that prenatal depression prevalence increases from the first to the third trimester of pregnancy which also contradict with previous study reports. What did the authors think for potential justification for this increment?

The authors also did not mention any possible reason, justification for their findings. I strongly recommend also the authors to state the possible justification for each contradicting findings and also the implication of the findings.

Page #10, last paragraph, I am not clear about this paragraph. The paragraphs describe about evidences taken from other source. The authors did not mentioned what happened on their review about social support. Did the authors assessed the effects of social support on prenatal depression? If so what was the relationship? On the same paragraphs last statement has no source?

Limitation

The author described somewhere else in the method section that they included articles that assessed prenatal depression using standard measured or standardized approach using validated screening tools. However, the authors mentioned measurement bias as a limitation. How could this go together? What did the authors did to overcome this limitation?

Reference

The authors describe somewhere else in the method section that they managed the reference using EndNote software. However, most references listed under the referral list is not correct and does not fulfill the journal requirement. Therefore, I strongly recommend the authors to revise the EndNote library and also correct the reference list. Example: look reference #2, consider others also.

Figure

Figure 2, delete the weight of individual /primary studies. What is PRR mean in the figure?

In forest plots; only surname of authors will be enough, and studies should be either in alphabetic order or in year of publication order. Revise all.

6. PLOS authors have the option to publish the peer review history of their article (what does this mean?). If published, this will include your full peer review and any attached files.

Reviewer #1: Yes: Pammla Petrucka

Reviewer #2: Yes: Cheru Tesema Leshargie

---

## [Author Response · Author response to Decision Letter 0]

27 Oct 2019

All issues raised by reviewers have been addressed and submitted in response to reviewers file.

---

## [Editor Report · Decision Letter 1]

11 Nov 2019

PONE-D-19-25076R1

Antenatal depression and its Effect on Birth Outcomes in Low and Middle-income Countries: A Systematic Review and Meta-analysis

PLOS ONE

Dear Mr Fekadu,

Thank you for submitting your manuscript to PLOS ONE. After careful consideration, we feel that it has merit but does not fully meet PLOS ONE’s publication criteria as it currently stands. Therefore, we invite you to submit a revised version of the manuscript that addresses the points raised during the review process.

We would appreciate receiving your revised manuscript by Dec 26 2019 11:59PM. To enhance the reproducibility of your results, we recommend that if applicable you deposit your laboratory protocols in protocols.io, where a protocol can be assigned its own identifier (DOI) such that it can be cited independently in the future. For instructions see: http://journals.plos.org/plosone/s/submission-guidelines#loc-laboratory-protocols

We look forward to receiving your revised manuscript.

Kind regards,

Animut Alebel, BSc, MSc

Academic Editor

PLOS ONE

Additional Editor Comments (if provided):

Editor’s comments

Title “Antenatal depression and its Effect on Birth Outcomes in Low and Middle-income Countries: A Systematic Review and Meta-analysis”

• Thank you for submitting your revised version of your manuscript. Most of the reviewers comments were addressed, but before going further steps, please address the following concerns.

Introduction

• Please update all old references used in the introduction section. For example, in 2015, the global prevalence of depression was estimated to be 4.4%.

Methods

• On your data sources (searching engine), why you did not search from EMBASE, which is a large database. Besides, you systematic review focused on depression. So, why not you searched from PsycINFO?

• Why you restricted your period from January 1, 2007 and December 31, 2017?

• I saw your search strategy from additional files, but I am not happy because it was not detail and compressive.

• Who are high risk?

• Could you attached your NOS tool because as far I know NOS has 10 maximum points?

• How you decided the category of quality scores as good, fair and poor?

• You reported a meta-analysis of proportions for antenatal depression. Therefore, for proportion or prevalence meta-analysis, the appropriate command is metaprop instead of metan command. Metan command is used to estimate the effect. Could you revise your analysis based on the above suggested command?

• For publication biases, objective tests (Egger's and Begg's tests). Therefore, please remove funnel plot.

• All the method components of PRIMA checklist are not reflected in the method section your review. Please include all the components listed in the method section of the PRISMA.

• Please include data extraction your method section.

• Risk of bias or quality score for each study should be assessed by two reviewers. Please say something about this. Moreover. If there were any disagreement during quality assessment, how this was resolved it? If quality assessment was done by two reviewers, and if there were any disagreements, please do KAPA test?

• Results

• The I2 value =96.7%, but in systematic review and meta-analysis, if the I2 > 75% it is better to report as a systematic review rather than merging heterogeneous results as a meta-analysis. How do you see this result?

• What is “the tooled” means?

• You tried to see the trend of depression over a time by dividing publication year. If you are interested to see the time trend in meta-analysis, I recommend you to do cumulative meta-analysis. Besides, to assess trend I recommend you to use study period rather than publication year.

• You included case control study to assess effect of antenatal depression on adverse birth outcomes. Can we assess effect using case control study?

• Discussion

• Please remove sub-headings from discussion.

• “This might associate with the number of included studies during different trimesters, the pooled prevalence was high where large number of studies included and low where small studies were included as adjustment was not made on the number of studies”. This is the limitation of this review rather than your justification.

• “We found that antenatal depression prevalence increases from the first to the third trimester of pregnancy”. Any association between antenatal depression and physiological changes during pregnancy?

• “We noted that institutional based studies reported higher prevalence relative to community-based studies. So, how do you justify this result?

• Limitations

• Only studies reported in the English language were included in your review. This is one limitation.

• What was your rational to exclude low quality papers?

• Conclusion

• Be consistent and conclude according to your result. Besides, the conclusion part of your abstract is not in line with the conclusion part of in the main document.

• Finally, I recommend you to edit the paper again there are some spelling and grammar problems.

---

## [Author Response · Author response to Decision Letter 1]

4 Dec 2019

I have addressed all the queries by editor and submitted all the manuscript contents with rebuttal letter and track changes.

---

## [Editor Report · Decision Letter 2]

18 Dec 2019

Antenatal depression and its Association with Adverse Birth Outcomes in Low and Middle-income Countries: A Systematic Review and Meta-analysis

PONE-D-19-25076R2

Dear Dr. Fekadu,

We are pleased to inform you that your manuscript has been judged scientifically suitable for publication and will be formally accepted for publication once it complies with all outstanding technical requirements.

With kind regards,

Animut Alebel, MSc, PhD student

Academic Editor

PLOS ONE
---

## [Editor Report · Acceptance letter]

20 Dec 2019

PONE-D-19-25076R2 

Antenatal depression and its Association with Adverse Birth Outcomes in Low and Middle-income Countries: A Systematic Review and Meta-analysis 

Dear Dr. Fekadu:

I am pleased to inform you that your manuscript has been deemed suitable for publication in PLOS ONE. Congratulations! Your manuscript is now with our production department. 

With kind regards,

on behalf of

Mr. Animut Alebel 

Academic Editor

PLOS ONE